# Carry-Over Effects of Climate Variability at Breeding and Non-Breeding Grounds on Spring Migration in the European Wren *Troglodytes troglodytes* at the Baltic Coast

**DOI:** 10.3390/ani13122015

**Published:** 2023-06-16

**Authors:** Ignacy Gołębiewski, Magdalena Remisiewicz

**Affiliations:** Bird Migration Research Station, Faculty of Biology, University of Gdańsk, Wita Stwosza 59, 80-308 Gdańsk, Poland; i.golebiewski.460@studms.ug.edu.pl

**Keywords:** climate change, spring phenology, short-distance migrants, European wren, *Troglodytes troglodytes*, MOI, NAO, SCAND

## Abstract

**Simple Summary:**

The phenology of avian migration adjusts to rapid changes in the climate in Europe. The North Atlantic Oscillation (NAO), which reflects weather patterns in western Europe, has been demonstrated to influence the timing of bird migration. The question is, how does climate in other regions of their wintering and breeding grounds in Europe influence migrants, which move between these areas? We aimed to determine carry-over effects of the Mediterranean Oscillation Index (MOI), a proxy for non-breeding conditions in the eastern Mediterranean, and the Scandinavian Pattern (SCAND), which reflects conditions at the breeding grounds, in combination with NAO, on the timing of the European wren’s short-distance spring migration through the Baltic Sea coast over 1982–2021. We showed that a combination of winter and spring MOI and NAO, as well as SCAND from the previous breeding season, explained the year-to-year variation in timing of the wren’s spring passage at the Baltic coast three to ten months later. Our study reveals that conditions the migrants encounter at wide non-breeding and breeding grounds during the preceding year have a carry-over effect on spring phenology of migrants in Europe, while spring conditions just fine-tune their timing.

**Abstract:**

Many studies have linked changes in avian phenology in Europe to the North Atlantic Oscillation (NAO), which serves as a proxy for conditions in western Europe. However, the effects of climate variation in other regions of Europe on the phenology of short-distance migrants with large non-breeding grounds remain unclear. We determined the combined influence of large-scale climate indices, NAO, the Mediterranean Oscillation Index (MOI), and the Scandinavian Pattern (SCAND), during the preceding year on spring migration timing of European wren at the southern Baltic coast during 1982–2021. We modelled the effects of these climate variables on the entire passage and subsequent percentiles of the wren’s passage at Bukowo-Kopań and Hel ringing stations. Over 1982–2021, the start and median of migration shifted earlier at Hel, but the end of passage shifted later at both stations. In effect, the duration of passage at Hel was extended by 7.6 days. Early passage at Hel was related with high MOI in spring and the preceding autumn. Spring passage at Bukowo-Kopań was delayed after high NAO in the previous breeding season, and high winter and spring NAO. Late spring passage occurred at both stations following a high SCAND in the previous summer. At both locations, an early start or median of passage followed high local temperatures. We conclude that phenology of the wren’s spring migration at the Baltic coast was shaped by conditions encountered at wintering quarters in western Europe, where NAO operates, and in the south-eastern Europe, where the MOI operates, in conjunction with conditions in Scandinavia during the previous breeding season. We demonstrated that climate variability in various parts of the migrants’ range has combined carry-over effects on in migrants’ phenology in Europe.

## 1. Introduction

Since the beginning of the twentieth century, the average global temperature has risen by more than 1 °C and continues to increase [1], affecting life cycles of the world’s flora and fauna at multiple levels. Birds are excellent indicators of the influence of climate change on animals [2] since they live in all climates, many species migrate seasonally, and many exhibit flexibility in response to environmental changes [3]. Many birds have been demonstrated to advance their spring migration in response to the changing climate and temperature increases at their breeding grounds [4], along migratory routes [5,6,7,8,9] or at wintering sites [10,11,12]. Numerous studies have linked such shifts in migrants’ phenology to climatic variability in the northern hemisphere, reflected by large-scale climate indices as the North Atlantic Oscillation Index (NAO) [13,14,15,16]. NAO has been used as a predictor of spring conditions in Europe, because it reflects temperature and precipitation [17], which are the key drivers of advancement of the breeding season in Europe [4,18]. Further, the Scandinavian Pattern (SCAND), which is a proxy for weather conditions across Scandinavia, has also been linked to migrant timing [19,20,21]. Large-scale climate indices in Africa, such as the Sahel Precipitation Index (SPI), Indian Ocean Dipole (IOD) or Southern Oscillation Index (SOI), have also been demonstrated to influence bird phenology, particularly in long-distance migrants that winter in areas where these indices operate [21,22,23,24,25]. However, the Mediterranean Oscillation Index (MOI), a predictor of climatic conditions in the Mediterranean region, where many European birds overwinter or stopover, has received little attention in the context of bird migrations. This is most likely because this climate index was only proposed in 1989, and its influence on climate in southern Europe is intertwined with that of NAO [26]. These large-scale indices reflect ecological conditions encountered by migrants at their breeding grounds, migration routes, or wintering areas, which may have carry-over effects on their subsequent life-stages [21,27,28]. In recent decades, many passerines have advanced the beginning and middle of their spring passage across Europe [9,20,29,30]. In certain European species, the residence period on the breeding grounds has extended due to a delayed autumn migration [28,29,31], which may favour more broods [28,32,33]. Short-distance migrants have shown the largest shifts to early spring arrivals in recent decades [15,16,17,34]. In addition, Forchhammer et al. [35] showed that conditions associated with NAO at the wintering grounds had a stronger influence on arrival in Norway for short-distance migrants than for long-distance migrants. Short-distance migrants use conditions at the wintering grounds as a cue for conditions at the breeding grounds, which are relatively close, in order to adjust their departure [17]. For this reason, larger fluctuations in migration phenology in Europe have been observed in this group than in long-distance migrants, for which conditions at their remote wintering grounds in Africa are poor predictors of the weather on their spring arrival in Europe [17,36,37]. Many studies have focused on the effect of climate variability on the phenology of long-distance passerine migrants [4,6,21,38,39,40]. Few studies, however, attempted to identify carry-over effects of conditions at breeding and non-breeding areas on migration timing in species with shorter migrations but larger wintering quarters in Europe [12,30,41]. To fill in this gap, we aimed to identify annual variation and any long-term trends in spring migration phenology over 40 years in a regular short-distance migrant, the European wren *Troglodytes troglodytes*, at the southern coast of the Baltic, Poland. We also aimed to determine any relationships between this species’ spring phenology and three large-scale indices that reflect climate across Europe, both at its breeding and non-breeding grounds.

## 2. Materials and Methods

### 2.1. Study Species

The European wren (henceforth wren) is a small insectivorous passerine common throughout Europe’s deciduous forests [42]. Populations breeding at the north and east of the species’ range are migratory, populations from western Europe are partially sedentary, and those breeding in the south, as well as endemic subspecies of Atlantic islands, are sedentary [43]. Our study focuses on northern populations, which usually migrate for 1000–1500 km, occasionally as far as about 2500 km, which we chose as an example of a regular short-distance migrant (Figure 1) [44]. Scandinavian breeding populations winter primarily in southern Sweden, central and western Europe, and only rarely on the Iberian Peninsula and in Great Britain (Figure 1) [44,45,46,47]. Wrens that migrate over the southern coast of the Baltic Sea in spring and autumn originate from Fennoscandia and the Baltic countries, with most ringing recoveries from Sweden. They overwinter mostly in western and central Europe, and in the Apennine Peninsula, but some ringing recoveries indicate that they also winter in the Balkans [44,48,49]. In Poland, including the Baltic coast, spring passage of wrens occurs in March–April [50]. The first individuals appear at the breeding grounds in southern Sweden in the second decade of April, and in northern Sweden from mid-May [45]. In Scandinavia, the breeding season begins in May and lasts until July (Figure 2) [45,46]. Wrens usually have one clutch, but can have two to three broods [43,46]. They leave the northern breeding grounds in August–September, reach non-breeding grounds by the end of October, and stay there until February–early March [43,45,46]. Thus, for the wren, we defined “spring migration” as March–April, “early breeding” as May–June, “late breeding” as July–August, “autumn migration” as September–October, and “non-breeding period” as November–February (Figure 2).

### 2.2. Study Sites and Material

We aimed to analyse the timing of spring migration of the wren at the southern coast of the Baltic Sea in 1982–2021. We used data from two ringing stations at the Polish coast (Figure 1): Bukowo-Kopań (54°20′13″ N, 16°14′36″ E) and Hel (54°44′25″ N, 18°33′49″ E). At both stations, birds were caught during spring migration (26 March–15 May), using a standardised monitoring protocol [52]. This study period covered the entire migration of wrens at the Polish coast. Migrants were captured daily, from dawn to dusk, using 8 m long mist nets. The number of mist nets at each station was stable throughout each season, but varied from 35 to 58 between across years. The numbers of caught European wrens, catching days and nets by year are presented in Table A1 in the Appendix A. Appendix A includes other additional information and results (Appendix A, Table A1, Table A2, Table A3, Table A4, Table A5, Table A6, Table A7, Table A8 and Table A9, Figure A1 and Figure A2). Following ringing, each bird was aged, measured and weighed, in accordance with the fieldwork protocol [52]. Sexing wrens in the field was impossible, due to a lack of sexual dimorphism in size or plumage [53,54]. Most adult and immature wrens look the same in spring, with only some immatures distinguishable by plumage [53,54]; thus, we combined all age groups in this study.

### 2.3. Climate Indices

We used 11 large-scale climate indices, which reflect weather conditions at summer breeding grounds (SCAND, NAO), spring migration routes (SCAND, NAO, MOI) and at the wintering areas (NAO, MOI) of wrens, and the local spring temperature at each ringing station, over 1981–2021 (Table 1). We averaged values of these indices for periods of months corresponding to subsequent stages in the wren’s year (Figure 2), and we used these averaged indices relevant to the regions where wrens should stay in each period. Thus, for example, we selected the Scandinavian Pattern (SCAND) from March to October (Figure 2), i.e., the periods when wrens are likely to stay within the influence of this climate pattern, but we excluded SCAND for November–December, when most wrens overwinter in southern Europe (Figure 1), away from the effect of SCAND. The large-scale climate indices use differences in the sea level pressure between two geographically distant weather stations to reflect weather patterns, which manifest across large areas in temperature, precipitation, wind direction and cloudiness [55]. The positive phase of SCAND in spring and summer (March–August) corresponds to high average temperatures and dry conditions across Scandinavia, and low temperatures and rainy conditions over central and western Europe and central Russia [19,55,56]. Spring and early summer SCAND exhibited a significant decreasing trend over our study period, but there was no long-time trend in late summer SCAND (Figure A1). Positive North Atlantic Oscillation Index (NAO) values are associated with warm winter (November–February) and spring (March–May) over western Europe, moist conditions between Iceland and Scandinavia, including western Europe, and dry conditions over Greenland, central and southern Europe [26]. A positive NAO in summer (June–August), is associated with dry and warm weather throughout northern Europe [57]. Late summer NAO had a significant decreasing trend over 1981–2021, although there were no clear trends in other periods, (Figure A1. Remisiewicz and Underhill [21] provided more details on the effects of these large-scale indices on long-distance migrant birds. The Mediterranean Oscillation Index (MOI), which we used as a proxy for conditions of the eastern Mediterranean region, was derived as the difference in the normalised air pressure at the sea level between Algiers and Cairo [58,59,60]. Positive values of winter and spring MOI are related with low rainfall, high temperatures, and hence mild winters and springs in the eastern Mediterranean region [26,60]. MOI showed no long-term trends for any of the months we studied (Figure A1).

Additionally, we used the mean daily temperatures averaged for spring (March–April) from the nearest weather stations as a proxy for local spring conditions at the ringing stations (Table 1). We used these temperatures as the monthly anomalies from the 1982–2020 baseline for March and April, which we then averaged for both months. For Bukowo-Kopań (54°20′13″ N, 16°14′36″ E) we used temperature anomalies averaged from two nearby weather stations at Koszalin (54°12′15″ N, 16°09′19″ E) and Łeba (54°45′12″ N, 17°32′04″ E) (TBK, Table 1). For the Hel ringing station, we used temperatures from the weather station in the Hel harbour (54°36′12″ N, 18°48′42″ E) (THL, Table 1). We used the monthly anomalies provided by Climate Explorer facility (https://climexp.knmi.nl/, accessed on 22 May 2023), which was based on data from the European Climate Assessment and Dataset (http://www.ecad.eu, accessed on 22 May 2023). These spring temperatures in Bukowo-Kopań and Hel increased by 1.2 °C and 1.1 °C, respectively, over 1982–2021 (Figure A1).

### 2.4. Statistical Analysis

We used the daily numbers and spring totals of wrens caught at each station, including only the first capture of an individual in a season (Appendix A, Table A1). For single days when mist-netting was suspended because of storms or other obstacles, the missing daily totals were imputed with the numbers estimated based on the numbers of wrens caught at this date during six earlier and six later years at that station, as in other studies [20,21,61,62]. Imputations constituted only 1–4 days in 9 seasons at Hel and in 13 seasons at Bukowo-Kopań. The analysis did not cover springs of 2011 and 2020 at Bukowo-Kopań, when this station did not operate, because of logistics or COVID–19 constraints. During all the other springs sufficient numbers of wrens were caught (Table A1). For Hel, years when less than 10 birds were caught during spring at a station were excluded from the study because the sample was too small for analyses. The reason for such small numbers of wrens caught in some springs might have been weather conditions that made them skip the stopover at Hel, as the numbers of mist nets and catching days were similar as in the other years (Table A1). Thus, we analysed data from 38 years at Bukowo-Kopań and 32 years at Hel, within the period 1982–2021 (Table A1).

For each station, based on daily catching totals, we calculated the annual anomaly (AA), which indicates the departure of the cumulative curve of the spring migration in one year from the many-year average curve (1982–2021), as in Remisiewicz and Underhill [20]. We calculated the AA for each spring, and compiled them in two time series, one for each station, which reflected the variation in overall spring migration timing over 1982–2021. To investigate the timing of subsequent phases of passage in more detail, we calculated the dates (Julian day) when the first 5% (beginning of passage), 50% (middle) and 95% (end of passage) of wrens occurred during every spring at each station, as in other studies [8,9,29]. In this way, for each station we obtained time series over 1982–2021 for four measures of migration timing: the AA, and the dates of 5%, 50%, and 95% of passage. We also calculated the duration of migration at each station as the difference between the dates for 5% and 95% of passage in each year. We compared these parameters of passage between stations over 1982–2021 using the Mann–Whitney test. We also used simple linear regression models to check for any significant trends in all the four measures of migration timing, and for migration duration, at each station over 1982–2021.

To explore the effects of climate indices on the wren’s passage at each station, we used all four measures as response variables for Bukowo-Kopań and Hel separately in multiple regression models against “Year” and 12 climate factors, which were the explanatory variables. These climate variables included eleven large-scale climate indices and one index of local temperature relevant for each station, TBK or THL, respectively (Table 1). The indices of climate in spring came from the same year as the analysed spring migration. However, for the climate indices that operate at wintering grounds, autumn migration routes and the breeding grounds, where wrens occur three to ten months prior to the spring migration in question, we used the data from the previous year (Table 1). All explanatory and response variables, which were in different scales, were standardised to have a mean of 0 and a standard deviation of 1, except for the “Year”, which was used as the year number since 1982. All climate variables were used in regression models in linear terms, as in previous studies [20,21,63].

Before running any multiple regression models, we used the Pearson correlation coefficient to check for any strong correlations between climate variables (|r| ≥ 0.7), which could cause multicollinearity [64]. For most climate variables, correlation coefficients, if ever significant, were below this threshold, but MOI and NAO in winter were correlated at r = 0.74 (Table A2). Thus, we ran the models without winter MOI, and then without winter NAO, and compared the obtained best models. The best models from these two approaches were the same for most response variables (compare Table 2 and Table A3). The only exceptions were the models for 50% at Bukowo, where the best model obtained when excluding winter MOI explained 40% of the variation (Table 2), more than the best model from approach excluding winter NAO (33%, Table A3). Thus, for 50% at Bukowo, we chose to present the best model from this first approach (Table 2).

We selected the best model based on the “all subsets regression” and ranking of the models by the Akaike information criteria adjusted for small samples (AICc) (Table A6, Table A7, Table A8 and Table A9). For the best models, with the lowest AICc, we calculated partial correlation coefficients (pR) to check the direction and strength of the correlation between the response variable and each explanatory variable, while excluding the effects of the other variables. The proportion of the variation explained by each best model was reflected by the adjusted coefficient of determination (AdjR^2^). We also calculated variance inflation factors (VIFs) for explanatory variables in each best model, to monitor for any effects of multicollinearity [64]. Statistical analyses were conducted in R 4.2.2 [65] and Statistica 13.3 [66].

## 3. Results

### 3.1. Long-Term Trends in Timing of Wren Spring Passage at the Southern Baltic Sea Coast

The overall spring migration (AA) of wrens showed a tendency to shift earlier by 1.7 days on average at Bukowo-Kopań (Figure 3A) and 2.7 days at Hel (Figure 3B) over the 40 years we studied, but these trends were not significant (Table A4). The timing of passage showed large year-to-year variation (Figure 3A,B), which was correlated between the stations (r_S_ = 0.42, *p* = 0.02). At Bukowo-Kopań, the start of passage (5%) was delayed by 0.02 days, the middle (50%) advanced on average by 5.3 days, but the end (95%) shifted later by 2.3 days (Figure 3C) over 1982–2021; thus, the duration of passage extended also by 2.3 days (Figure A2), but none of these trends was significant (Table A4). At Hel, the start of the wren’s passage shifted earlier by 3.6 days, the middle by 5.0 days, but the end of passage shifted later by 3.9 days over the 40 years of our study. These trends were not significant (Figure 3D, Table A4), but the duration of spring passage at Hel extended significantly (*t* = 2.20, *p* = 0.04) on average by 7.6 days, due to the advance of the initial phase of passage and the delay of its final phase (Table A4, Figure A2). The dates of 5%, 50% and 95% of the passage were on average 2–4 days later at Hel than at Bukowo-Kopań (Table A5), and the difference was the largest for the start (5%) of the passage (Mann–Whitney test, Z = 6.24, *p* < 0.01).

### 3.2. Climate Indices and the Timing of Spring Passage

At Bukowo-Kopań, the overall spring passage (AA) was later with higher values of NAO in the late summer of the preceding breeding period, which explained AdjR^2^ = 15% of the variation (Table 2, Figure 4A). 

However, MOI was not selected for in any best model, in contrast to Hel (Table 2). At Bukowo-Kopań, the timing of the first phase of the passage (5%), analogously as the whole passage, was later with a higher NAO and SCAND in the late summer of the previous year, which jointly explained AdjR^2^ = 28% of its variation (Table 2). The median date of passage (50%) was later with a higher early summer SCAND and a late summer NAO of the previous year, and the winter and spring NAO, but earlier with a higher spring MOI, SCAND and local temperatures during migration in March–April (Table 2, Figure 4A). The best model including all these climate variables explained AdjR^2^ = 40% of the variation (Table 2). The end (95%) of the passage was later with a high NAO in the late summer and with a low SCAND in the late summer of the previous year; this model explained only AdjR^2^ = 10% of the variation (Table 2, Figure 4A).

At Hel, the overall timing of the passage was earlier with a higher spring MOI, and with a lower late summer SCAND in the previous year; this best model explained AdjR^2^ = 16% of the variation of the AA, but did not include NAO (Table 2, Figure 4B). The initial phase of the passage (5%) was earlier with higher spring local temperatures at Hel, and the previous last years’ autumn MOI, and with a lower early summer SCAND in the early summer of the previous year; these factors jointly explained AdjR^2^ = 39% of the variation. The median date of passage (50%) was later the higher the last years’ late summer SCAND, which explained AdjR^2^ = 14% of the variation. The end of the passage (95%) was earlier with a high spring MOI, but this model explained only AdjR^2^ = 1% of the variation (Table 2, Figure 4B). All VIFs < 10 in all the best models (Table 2) indicated no collinearity between the selected variables.

## 4. Discussion

We revealed a large year-to-year variation in the wren’s spring migration at the southern Baltic coast, but no long-term trends for any advance or delay of passage. Spring migration at the more western Bukowo-Kopań station was shaped mainly by the NAO during the previous breeding period, and by the winter and spring NAO. In contrast, the passage of wrens at Hel responded mostly to the spring and autumn MOI, but not to the NAO. We will suggest an explanation for these differences in responses of the wren’s passage to these climate factors at two coastal stations located only 160 km apart in the context of results from other ringing stations in northern Europe. At both stations, late spring passage occurred after a high SCAND during the breeding season. The beginning of the passage at Hel and the median at Bukowo-Kopań were early with high local spring temperatures. We will discuss how these climate factors, which operate mostly at breeding and wintering grounds, might have a carry-over effect on the wren’s spring migration timing at the Baltic coast three to ten months later.

### 4.1. Multi-Year Trends in the Timing of Wren Spring Migration

Many European passerines [3,16,29,67,68,69], as well as North American species [9,70,71], including the house wren *Troglodytes aedon* [72], have advanced their spring migration timing. Analogous shifts to earlier spring arrivals have also occurred in the European wren, based on data from other parts of the Baltic region [8,73], and from the Helgoland island in the south-eastern part of the North Sea [14]. The mean date of the wren’s spring passage advanced significantly at Helgoland over 1960–2000 [14], but we found only small advances in the median date of passage over 1982–2021 at both our stations, located 550–700 km to the east (Figure 1). We revealed a slight advance of the passage of wrens on the southern Baltic coast, similar to that found at Ottenby station on the Baltic island of Öland (Figure 1) in 1971–2002 [69]. However, our results differ from those at Christiansø island (Figure 1), where both the beginning (5%) and the end (95%) of the wren’s passage advanced significantly over 1976–1997 [8], but in our study the end of passage was slightly delayed. In our study, the most pronounced advance was that of the median (50%) of migration, but at Christiansø, the shift of the median was the least pronounced shift of all the percentiles. These discrepancies can be explained in several ways. Firstly, the study period at our stations is more recent than the periods analysed for the other stations [8,14,69,73]. Many-year trends may differ depending on the range of years selected for analyses, which would include various combinations of early and late migrations, as demonstrated for the migration of the song thrush *Turdus philomelos* in the Baltic region [30,62]. Considering the large inter-annual variation in the wren’s passage (Figure 3), late passage in several springs within 2011–2021 likely caused a lack of clear trends for earlier arrivals, in contrast to trends from earlier periods at other stations. Secondly, proportions of various migratory populations, each with its own multi-year trends and fluctuations, are likely to differ between distant stopover locations, and our two stations are farther east than Christiansø [8] and the other compared sites (Figure 1). Thirdly, birds intensively refuel before crossing ecological barriers, such as the open sea. This can cause mixing of different populations at stopover sites, and may affect the timing of their further migration, resulting in discrepancies between coastal and island sites [74,75,76]. Further, adults and immatures differ in their propensity to cross ecological barriers, which leads to the “coastal effect”, i.e., a greater proportion of immatures observed at the coast than inland [77]. Thus, we suspect that different proportions of age groups may cause differences in the timing of the passage between the coastal (Bukowo-Kopań, Hel) and the island (Christiansø, Ottenby) stations in the Baltic region. However, given that ageing of wrens is problematic in spring [53,54], we lack clear evidence to support this assumption.

The later occurrence of the corresponding phases of passage at Hel by a few days compared to Bukowo-Kopań may reflect the time it takes wrens to cross the extra ca. 160 km to the west to reach Hel (Figure 1), considering they migrate slowly [76]. An analogous time difference in spring migration at these two stations was found for the European robin *Erithacus rubecula* [78,79]. However, this shift in migration timing might also be a result of different proportions of various migratory populations of wrens passing through both stations, which we will discuss later.

The significantly extended duration of passage at Hel was due to the delayed end of migration, which did not occur at Bukowo-Kopań. A tendency towards an extended duration of the spring passage is uncommon among European migratory birds. However, some long-distance migrants, such as pied flycatcher *Ficedula hypoleuca* and red-backed shrike *Lanius collurio,* showed a tendency to advance the beginning and delay the end of the spring migration, and thus to a prolonged passage at Christiansø [8]. For the song thrush *Turdus philomelos*, which has a similar migration distance and routes to the wren [62], the end of the spring passage at the Baltic coast of Poland was delayed after a warm December–February at their wintering grounds in south-western Europe [30]. We suspect that wrens have a similar mechanism for delaying the end of the passage in response to conditions at their wintering grounds.

### 4.2. Effects of Conditions on Spring Migration Routes on the Timing of Spring Passage at the Baltic Coast

The earlier migration of wrens with higher local spring temperatures we found at both stations (Table 2, Figure 4A,B) is consistent with the overall trend of migrants to shift spring migration earlier in response to increasing temperatures in Europe [6,9,14,15,17]. This trend mainly considers short-distance migrants [15,17,23], including our study species. With high local temperatures, vegetation is early, allowing for efficient foraging, which, in turn, may shorten the stopover duration [35,80], thus the passage through subsequent stations occurs earlier and quicker. Birds’ mobility will also be greater during warm conditions and the abundance of insects than on cold and rainy days, increasing the likelihood of catching them in mist nets [81]. Local temperatures at Bukowo-Kopań (TBK) and Hel (THL) are also related to the conditions birds encounter earlier on their spring migration routes, as temperatures are correlated across larger regions of Europe [82,83]. The timing of arrivals of the first 5% of wrens at Hel (Figure 4B) and 50% at Bukowo-Kopań (Figure 4A) were related to local temperatures, as in other studies [8,23]. However, the overall migration (AA) at both stations was related rather to the indices of climate at earlier stages of the wren’s spring migration, and in the previous breeding season, implying that, as in other species, local spring temperatures only fine-tune the timing of passage [17,21].

Spring passage of wrens at the southern coast of Baltic was earlier with higher spring MOI, which reflects warm and dry weather in the eastern Mediterranean area [26,59]. Such conditions would promote early and fast passage farther north, and thus early arrivals at the Baltic coast. The overall migration and its end for the wren at Hel were early with a high spring MOI (Figure 4A), and thus a warm spring on the route from the eastern Mediterranean. Such conditions would create favourable feeding conditions on stopovers, allowing even the youngest and weakest individuals, which migrate last, to arrive relatively early at the Baltic coast [76,84]. Analogously, warm and dry conditions in the eastern Mediterranean would explain the relation of an early median of passage through Bukowo-Kopań with a high spring MOI (Figure 4B).

The spring MOI reflects the climate over the eastern Mediterranean region, but less so in western Europe [26] which, in the spring, is rather under climatic influence of NAO. In the spring, these indices were not correlated (Table A2). The stronger influence of the spring MOI on the overall passage at Hel than at Bukowo-Kopań corresponds with the pattern of ringing recoveries of wrens, implying that Hel supports a greater proportion of individuals wintering in the eastern Mediterranean region than Bukowo-Kopań, where the populations wintering in south-western Europe seem to prevail [44,48,49]. This pattern is consistent with the relationship between the spring NAO and the median date of passage at Bukowo-Kopań, but not with the migration timing at Hel (Figure 4). Surprisingly, a high spring NAO was related with a late median of passage at Bukowo-Kopań (Figure 4). The positive phase of the NAO brings warm and wet weather to most of continental Europe [85]. Some nocturnal species, like thrushes, avoid leaving stopover sites during rainfall [86], thus such conditions on migration route may also delay wren migration.

The positive phase of spring SCAND was related with the late beginning of the passage at Hel (Figure 4B). A positive SCAND in winter and early spring results in low temperatures in western Europe and high precipitation over central and southern Europe [19]. These conditions might be unfavourable for passage [87], delaying wrens’ departures from these areas, and thus their arrival at Hel. As the spring progresses, the centre of influence of the positive phase of the SCAND shifts southward, bringing warm and dry conditions to western Europe [88]. This change might explain the relation of an early median of wren migration at Bukowo-Kopań with a high SCAND in spring (Figure 4A).

### 4.3. Effects of Conditions on Wintering Grounds on the Timing of Spring Passage

The winter NAO has been shown to influence the spring migration of many European short-distance migrants, such as the goldcrest *Regulus regulus*, song thrush *Turdus philomelos* or skylark *Alauda arvensis*, as well as long-distance migrants, such as the willow warbler *Phylloscopus trochilus* or the lesser whitethroat *Curruca curruca* [10,35,62]. With a positive winter NAO, which indicates mild and wet winters and early spring in western and central Europe [14,69,89], the majority of these passerines migrated early in spring at various locations throughout Europe [14,21,69]. Favourable conditions at wintering grounds improve the fitness of migrants, which enables them to depart early and reach the breeding grounds quickly in spring [14,90]. Mild winters may also cause a shortening of migration distance, and thus early arrivals in the north, as suitable wintering conditions may occur closer to breeding grounds than during cold winters [10,91]. In contrast to these studies, which emphasize early spring migration of the wren after a positive winter NAO [14,69], we found that the median date (50%) of passage at Bukowo-Kopań was later with a higher winter NAO. The explanation might be that mild winter conditions, associated with a positive winter NAO, may favour the survival of inexperienced and weak individuals, particularly from late broods [43]. The increased proportion of immatures and weak individuals that survived a mild winter well and arrived at Bukowo-Kopań later than the fittest birds may delay the bulk of the passage and thus shift the median later, as in the song thrush at the Baltic coast [30]. This influence of the winter NAO suggests that the wintering grounds of wren populations migrating through Bukowo-Kopań are primarily in south-western Europe. We found no analogous influence of the winter NAO on the wren’s passage at Hel, which implies that populations passing through that station are likely to winter mainly outside the influence of the NAO [14,69], most likely in the more south-easterly part of Europe, where the MOI shapes the climate [26,60].

### 4.4. Carry-Over Effects of Conditions at the Breeding Grounds on the Following Spring Passage

We found that with positive values of indices of climate at the breeding grounds in Scandinavia, such as the NAO and SCAND in early and late summer, the overall migration, and most phases of the wren’s passage, were late in the spring of the next year at both our stations. The only exception was an early end (95%) of the passage at Hel with a positive SCAND in the late summer. The SCAND is strongly associated with the conditions in early summer [19], which is the time of the first and second broods in wren [43]. The NAO has the greatest influence on weather conditions in late summer in north-western and central Europe, during the second and third broods [43], but has the least influence on the weather in June [57]. Our results are consistent with these patterns, as the SCAND in early and late summer was related to the timing of the passage the next spring at both of our stations (Figure 4A,B).

High pressure in continental Europe and the Azores, correlated with a positive NAO, and the centre of air circulation occurring over Scandinavia in the summer [21], are both associated with high temperatures and low precipitation at wrens’ breeding grounds in the Baltic region and further east. High SCAND and NAO in summer would thus favour early breeding and provide suitable conditions for wrens to have additional broods, increasing the numbers of juveniles [18,92] and, thus, the abundance of their migration next spring [93]. Young from early broods have more time to mature and prepare for autumn migration than those from later broods, and thus have a better chance of claiming a winter territory, which would further increase their survival rate [94,95]. This might result in numerous arrivals of these individuals in the springs following summers with a high NAO and SCAND (Figure 4A,B). Immatures usually migrate later in the spring than the more experienced adults [87]. Thus, numerous spring returns of young after warm summers would result in later timing of overall migration, and of its subsequent phases, than in springs after cold summers and fewer immatures, as in the song thrush [30]. Favourable breeding conditions may improve immature and adult fitness [4], which would enable some individuals to reach farther wintering grounds, from where they will arrive later than after cold and wet summers. At Hel, the late summer SCAND had a significant effect on the spring passage, but we found no influence of the summer NAO (Figure 4B). In contrast, the late summer NAO had the greatest influence of all climate variables on the spring passage at Bukowo-Kopań (Figure 4A), but the early and late summer SCAND also had an effect on the subsequent phases of the passage at that station (Figure 4B). The influence of the summer NAO on migration through Bukowo-Kopań, but not at Hel, suggests that populations passing through these stations encounter different breeding conditions, and thus their breeding ranges partly differ. This corresponds with the distribution of ringing recoveries, which suggests that, at Bukowo-Kopań, a greater proportion of wrens originate from north-western breeding grounds, such as Sweden and Norway (Figure 1), where the NAO operates, than at Hel, which supports more birds breeding to the east, under greater climatic influence of the SCAND [44,48,49].

The late median of passage at Bukowo-Kopań with a positive SCAND in a previous early summer (Figure 4A), can be explained by young from early broods staying on the breeding grounds longer due to an early and warm summer. Such good conditions may allow them to adequately prepare for the autumn migration. As a consequence, some immatures might be able to migrate farther, and thus return in spring later than after unfavourable summers. The late end of the passage at Bukowo-Kopań after positive NAO in late summer (Figure 4A) might be due to an increased proportion of young from second broods arriving at the Baltic coast late the next spring [96].

On the other hand, the end of migration at Bukowo-Kopań was early after positive SCAND in late summer. We consider that the last cohorts of migrants may have included some proportion of a population originating from eastern Europe, where a positive summer SCAND means low temperatures and high precipitation [19,55,56]. It could have the opposite effect to the influence of the late summer NAO, causing fewer broods, and low fitness and survival rate of immatures. However, given that conditions at the breeding grounds explained only 10% of the variation in dates of 95% at Bukowo-Kopań (Table 2, Figure 4A), we approach these results with caution.

### 4.5. Carry-Over Effects of Conditions on Autumn Migration on the Following Spring Passage

We found that, when the autumn MOI was high, and thus the eastern Mediterranean region experienced high temperatures and low precipitation [60,96], the arrivals at Hel began early in the following spring (Figure 4B). Given recently observed changes in migratory behaviour and great plasticity of short-distance European migrants [7,97,98], we consider that shortening of migration distance is a plausible explanation. Mild autumn conditions may cause some individuals to overwinter closer to the breeding grounds [91,99,100,101], or to skip migration and stay in the breeding area for the winter [102,103]. With favourable conditions along the autumn migration route, wrens may spend winter closer to breeding grounds, e.g., in the Apennine Peninsula, without crossing the Mediterranean Sea, resulting in their early spring return to the north. Shorter distances between the wintering and breeding sites would benefit wrens in several ways. Birds staying closer to breeding grounds would be able to predict conditions there more precisely than those staying farther away [3]. Because wrens have high energy demands during migratory flight, due to their short and rounded wings and low rate of fat deposition [76], a shorter distance to cross would improve their survival of both autumn and spring migration. Closer wintering grounds might enable males to establish breeding territories early [80,104,105]. Females and juveniles, according to the dominance theory and given the greater physiological constraints [106], will be more likely to migrate farther south to overwinter in conditions that will ensure their survival [107,108]. Such patterns have been observed in the European robin, which is territorial at the wintering grounds, similarly to the wren [103], and its close North American relative the marsh wren *Cistothorus palustris* [106]. Shortening of the migration distance has been observed among passerines in response to climate change [87,100], such as the blackcap *Sylvia atricapilla* [99,100,101] or the blue tit *Cyanistes caeruleus* [109].

We also suspect another explanation for the wren’s early arrivals at Hel after warm autumns in the eastern Mediterranean region. Conditions at autumn stopover sites are crucial for the rate of migration, as birds spend up to seven times as much time on foraging and resting as they do on flying [110]. The efficiency of energy accumulation depends indirectly on conditions at stopover sites [87,111]. Warm autumn weather promotes robust vegetation and the availability of insects for migrating birds [20,82]. The wren’s diet is composed exclusively of invertebrates [112]. Increased availability of such prey in autumn may benefit their condition and allow for early arrivals at the Baltic coast the following spring. An analogous positive carry-over effect of favourable conditions on autumn stopover sites in the Mediterranean region on survival, and subsequent early arrivals at the breeding grounds, has been shown in the great reed warblers *Acrocephalus arundinaceus* [33]. A lack of analogous effects of the autumn MOI on the spring migration at Bukowo-Kopań could be attributed to a lower proportion of wrens arriving at that station from the eastern Mediterranean region than at Hel, which we discussed earlier [44,48].

### 4.6. Long-Term Trends and Inter-Annual Variation in Wren Spring Migration in Response to Trends and Variability in the Climate Indices

Our results revealed significant negative trends over 1982–2021 for the spring and early summer SCAND, and for the late summer NAO (Figure A1), indices related with spring and summer conditions in northern Europe [19,57]. Because the early summer SCAND and the late summer NAO were both positively associated with the passage at Bukowo-Kopań (Figure 4A), their combined linear multi-year trends might explain the slight shift to a later end (95%) of passage at this station (Table 2, Figure 4A,B). Analogously, the small advance in the beginning of passage at Hel (Figure 3D) could be due to the positively correlated spring SCAND (Figure 4B), which showed long-term trends. The influence of the local spring temperature at Hel (THL), which increased during our study period (Figure A1), may have contributed to that small advance in the start of the passage at Hel (Figure 3D), as in other passerines [21]. Local spring temperatures near Bukowo-Kopań (TBK), which increased over 1981–2021 (Figure A1), might also have contributed to the slight advance of the median date at this station (Figure 3C). These advances of the early phases of the wren’s passage at our stations correspond with the general trend in short- and medium-distance migrants in Europe to shift the beginning of their passage earlier with higher local spring temperatures [14,15,17].

The delay of the end of migration, which was the main cause of the extension of the spring passage over the 40 studied years at Hel, cannot be attributed to the effect of the spring MOI (Figure 4B), which had no trend over study period (Figure A1). Hence, other factors, most likely those related to changes in the number of broods and population productivity over the study period, which we discussed earlier, provide a better explanation for the prolonged passage at Hel (Figure A2, Table A4). An increase in the number of inexperienced and late-migrating immatures after warm summers that allow for two or three broods would delay the end of migration.

The insignificant trends, with a large year-to-year variation in the timing of the spring passage at both stations (Figure 3A,B), could be explained by a large interannual variation in the remaining climate indices related to the wren’s passage. Since the 1980s, the NAO and MOI have exhibited large year-to-year variability (Figure A1) [26,67,88]. The summer SCAND has shown a mostly negative linear trend since the mid-20th century [113], but its inter-annual variation increased since the 1990s [114]; thus, this trend in the late summer SCAND is no longer valid (Figure A1). A lack of significant long-term trend in the overall passage (AA) at Hel (Figure 3A), which was influenced by the spring MOI and the late summer SCAND in the previous year (Figure 4B), is consistent with no long-term trends in both these climate indices (Figure A1). Analogously, a large variation in the overall passage (AA) (Figure 3B), and in the dates of the beginning and the end of passage at Hel (Figure 3D) is consistent with large multi-annual variations in the spring and the previous autumn MOI (Figure A1). The winter and spring NAO, which showed large annual changes, but no long-term trends (Figure A1), may have contributed to the large variation, and thus a weak trend in the median date of passage at Bukowo-Kopań.

## 5. Conclusions

Our results provide new insight into the drivers of spring migration phenology in short-distance migrants with extensive non-breeding grounds, such as the wren, by considering a diverse set of climate indices cover most of Europe. The timing of the wren’s spring passage at the Baltic coast was shaped by a combination of carry-over effects of climate factors that the birds encountered in different parts of their range during the year prior to the spring migration. A high spring MOI and SCAND promoted early spring arrivals of wrens at the Baltic coast. Favourable conditions during autumn migration to the eastern Mediterranean, reflected by a high autumn MOI, likely advanced arrivals at Hel the next spring, allowing wrens to overwinter closer to the breeding grounds. Warm summers on breeding grounds, as indicated by a high summer SCAND and NAO, and mild winters in south-western Europe, as indicated by a high winter NAO, delayed the next years’ spring migration, probably as a result of the positive effect on the number and survival rate of immatures. These large-scale indices of climate in regions where wrens stayed during non-breeding and breeding seasons had greater impact on their spring phenology than the local spring temperatures, which only fine-tuned their migration timing. We attribute the weak multi-year trends in the timing of wren spring passage at both our stations, which differ from advanced arrivals at other sites in Europe, to the effects of climate indices with large year-to-year variations, such as the MOI, which likely outweigh the influence of the climate indices with long-term trends. Strong effects of the MOI on wrens’ migration at the more eastern station Hel emphasize that, besides the most commonly studied NAO, other large-scale climate indices shape spring migration phenology, especially in central and eastern Europe. The effect of the MOI on wrens’ spring migration highlights the relevance of this climate index, which has been largely overlooked, in the understanding of the effects of climate change on the phenology of bird migration in Europe.

## Figures and Tables

**Figure 1 animals-13-02015-f001:**
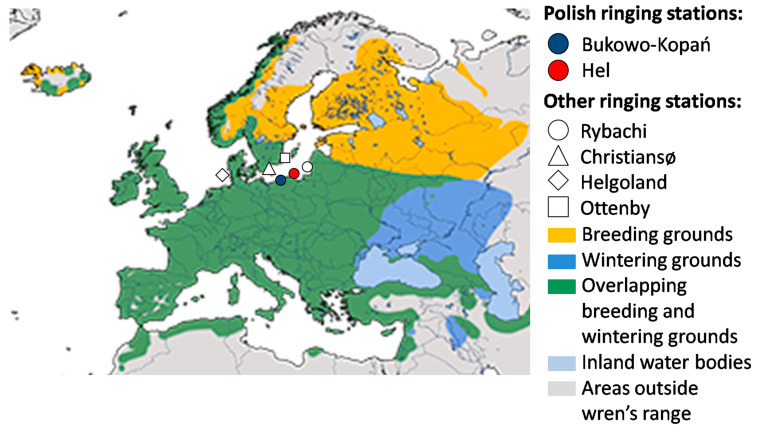
Geographical range of the European wren, two ringing stations on the southern Baltic coast in Poland, where the data we analyse was collected, and other ringing stations mentioned in the text (map after Trepte 2022 [51], modified).

**Figure 2 animals-13-02015-f002:**
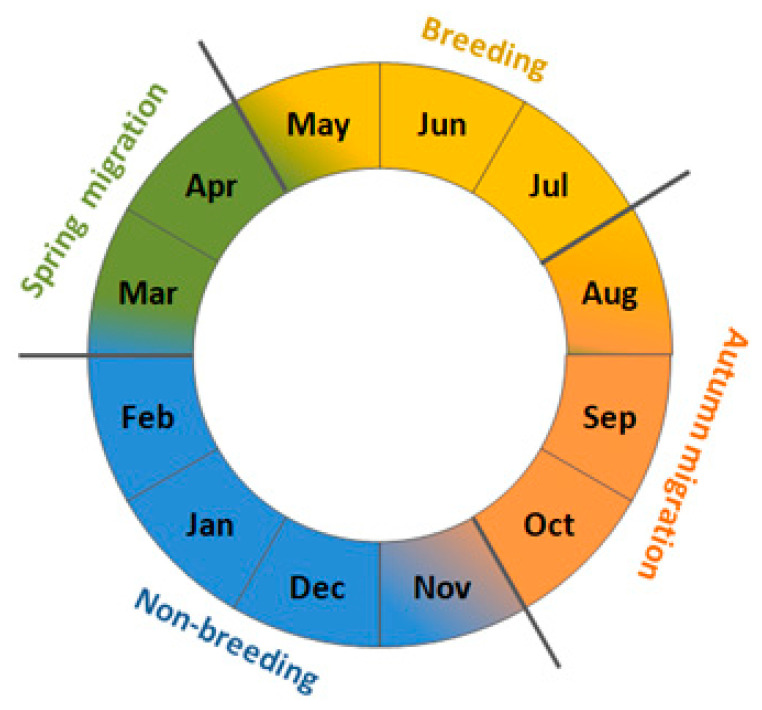
The four main periods of the European wren’s life cycle we used in analyses. Areas marked with transition between two colours indicate the possibility of life stages overlapping.

**Figure 3 animals-13-02015-f003:**
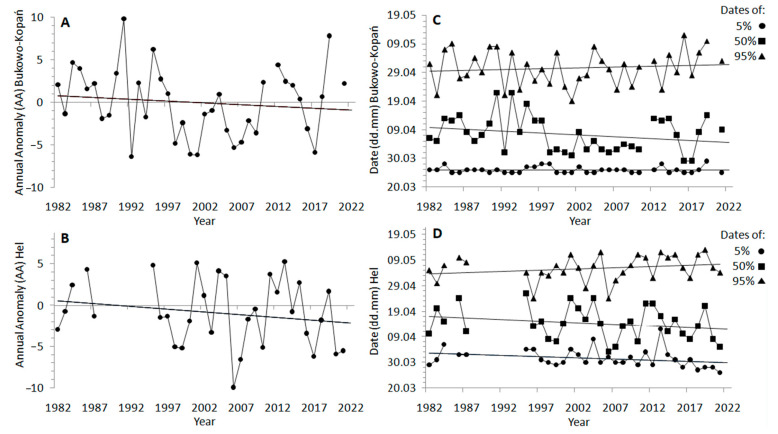
Multiple-year trends in the spring migration of the European wren over 1982–2021, based on the annual anomaly (AA) of passage at (**A**) Bukowo-Kopań and (**B**) Hel, and based on 5%, 50%, and 95% percentiles of passage at (**C**) Bukowo-Kopań and (**D**) Hel. Statistics of the regression equations in Table A4.

**Figure 4 animals-13-02015-f004:**
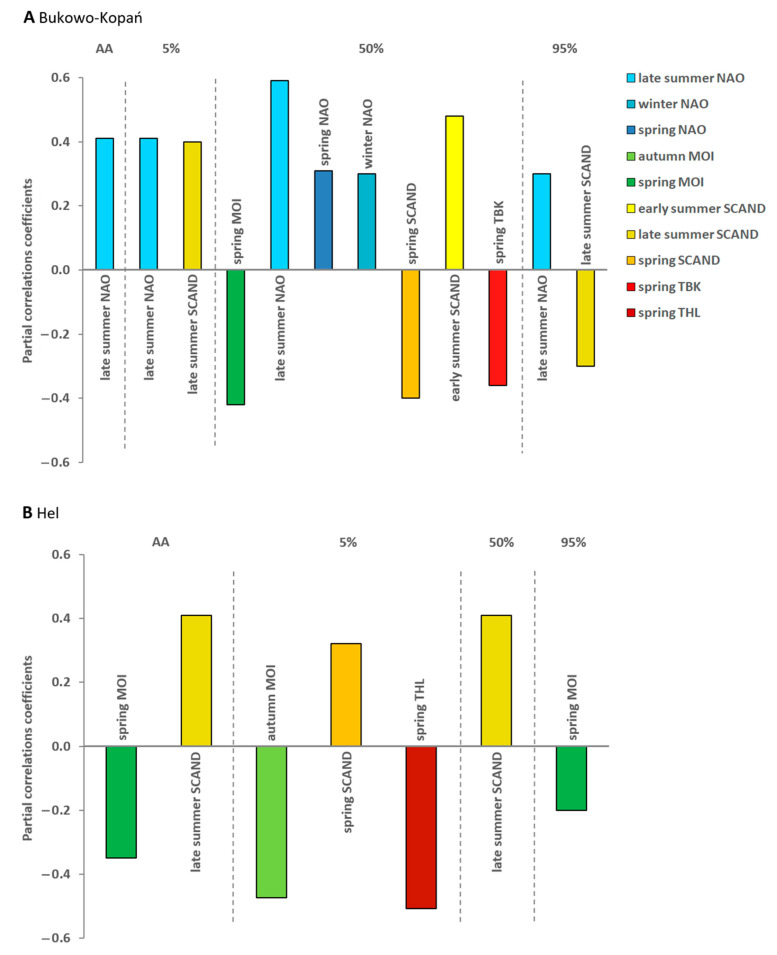
Partial correlation coefficients (pR) for the climate variables selected in the best models for the annual anomaly (AA) and selected percentiles (5%, 50%, 95%) of spring passage of the European wrens ringed in 1982–2021 in northern Poland, (**A**) at Bukowo-Kopań, and (**B**) at Hel, against the climate indices selected in the best multiple regression models. The negative value of the partial correlation coefficient (pR) indicates earlier arrival with a higher value of a climate index, the positive value means the opposite. The details of these models are presented in Table 2.

**Table 1 animals-13-02015-t001:** Climate variables and the “Year” used as explanatory variables in modelling the European wren’s spring migration (26 March–15 May) over 1982–2021 at Bukowo-Kopań and Hel ringing stations, northern Poland. Months = months corresponding with a life stage of the wren (Figure 2), for which the climate index was averaged, “_1Y” = months of the year prior to the analysed spring migration.

No	Abbreviation	Climate Index	Months	Source of Data
1	early summer NAO	North Atlantic Oscillation Index	May–Jun_1Y	https://www.cpc.ncep.noaa.gov/products/precip/CWlink/pna/nao.shtml (accessed on 22 May 2023)
2	late summer NAO		Jul–Aug_1Y	
3	autumn NAO		Sep–Oct_1Y	
4	winter NAO		Nov_1Y–Feb	
5	spring NAO		Mar–Apr	
6	autumn MOI	Mediterranean Oscillation Index (Algiers–Cairo)	Sep–Oct_1Y	https://crudata.uea.ac.uk/cru/data/moi/ (accessed on 22 May 2023)
7	winter MOI		Nov_1Y–Feb	
8	spring MOI		Mar–Apr	
9	early summer SCAND	Scandinavian Pattern Index	May–Jun_1Y	https://ftp.cpc.ncep.noaa.gov/wd52dg/data/indices/scand_index.tim (accessed on 22 May 2023)
10	late summer SCAND		Jul–Aug_1Y	
11	spring SCAND		Mar–Apr	
12	spring TBK	Mean temperature anomaly for Koszalin and Łeba (near Bukowo-Kopań)	Mar–Apr	http://www.ecad.eu/ https://climexp.knmi.nl/ (accessed on 22 May 2023)
13	spring THL	Mean temperature anomaly at Hel		
14	Year		1982 = year 1	Our data

**Table 2 animals-13-02015-t002:** Relationships, as shown by the best models, between the climate variables and the annual anomaly (AA), and selected percentiles (5%, 50%, 95%) of the spring passage of European wrens ringed at Bukowo-Kopań (BK) and Hel, northern Poland, in 1982–2021. Estimate = coefficients from multiple regression, SE = standard error of the estimates; *t*, *p* = *t*-test and significance of each estimate, *p* < 0.05 marked in bold, VIF = variance inflation factor. *R*^2^ = partial determination coefficients, *pR* = Pearson’s partial correlation coefficient.

Station/ Response Variable	Model Statistics	Explanatory Variable	Estimate	SE	*t*	*p*	VIF	*R* ^2^	*pR*
Bukowo-Kopań									
BK_AA	AdjR^2^ = 15%, F_1,37_ = 7.79	**late summer NAO**	0.42	0.15	2.79	**0.01**	1.00	0.17	0.41
BK_5%	AdjR^2^ = 28%, F_2,36_ = 8.23	**late summer NAO**	0.38	0.14	2.69	**0.01**	1.02	0.17	0.41
		**late summer SCAND**	0.37	0.14	2.63	**0.01**	1.02	0.16	0.40
BK_50%	AdjR^2^ = 40%, F_7,31_ = 4.69	**spring MOI**	−0.41	0.16	−2.58	**0.01**	1.61	0.18	−0.42
		**late summer NAO**	0.56	0.14	4.04	**<0.00**	1.21	0.34	0.59
		spring NAO	0.29	0.16	1.83	0.08	1.63	0.10	0.31
		winter NAO	0.27	0.15	1.77	0.09	1.48	0.09	0.30
		**spring SCAND**	−0.38	0.16	−2.41	**0.02**	1.62	0.16	−0.40
		**early summer SCAND**	0.46	0.15	3.03	**<0.00**	1.44	0.23	0.48
		**spring TBK**	−0.33	0.15	−2.15	**0.04**	1.49	0.13	−0.36
BK_95%	AdjR^2^ = 10%, F_2,36_ = 3.16	late summer NAO	0.30	0.16	1.91	0.06	1.02	0.09	0.30
		late summer SCAND	−0.29	0.16	−1.88	0.07	1.02	0.09	−0.30
Hel									
HL_AA	AdjR^2^ = 16%, F_2,30_ = 4.02	**spring MOI**	−0.35	0.17	−2.08	**0.05**	1.09	0.13	−0.35
		**late summer SCAND**	0.41	0.17	2.45	**0.02**	1.09	0.17	0.41
HL_5%	AdjR^2^ = 39%, F_3,29_ = 6.30	**autumn MOI**	−0.44	0.15	−2.90	**0.01**	1.11	0.22	−0.47
		spring SCAND	0.28	0.15	1.82	0.08	1.10	0.10	0.32
		**spring THL**	−0.48	0.15	−3.17	**<0.00**	1.09	0.26	−0.51
HL_50%	AdjR^2^ = 14%, F_1,31_ = 6.19	**late summer SCAND**	0.41	0.16	2.49	**0.02**	1.00	0.17	0.41
HL_95%	AdjR^2^ = 1%, F_1,31_ = 1.33	spring MOI	−0.20	0.18	−1.15	0.26	1.00	0.04	−0.20

## Data Availability

Data supporting reported results can be found at the Global Biodiversity Information Facility database at: Ringing Data from the Bird Migration Research Station, University of Gdańsk (gbif.org) [49] (accessed on 22 May 2023).

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
