# Peer review of "Carry-Over Effects of Climate Variability at Breeding and Non-Breeding Grounds on Spring Migration in the European Wren Troglodytes troglodytes at the Baltic Coast"

_animals, 2023, doi:10.3390/ani13122015_

Round 1

Reviewer 1 Report

The authors present phenology data of the European Wren from two Operation Baltic ringing stations. They correlate this data with meteorological patterns that affect the region and possibly influence the spring passage of the Wrens to the breeding grounds.  The study is well planned, data critically analyzed, and the paper on the whole is well written. There are however editorial glitches that need to be corrected. The paper needs to be proof-read either by a native English speaker, or an appropriate program/application.

The title is very long and cumbersome. Recommend to shorten to: Climate variability at breeding and non-breeding grounds have has stronger carry-over effects on spring migration in the European Wren Troglodytes troglodytes at the Baltic coast than spring conditions

Please desist from slang like “we set out” – used more than once!

Figure 1 – the Falsterbo square is not where it should be on the map

Figure 3 – suggest changing many by multiple

Line 556 – change Irends with Trends

Line 557 - change Rrends with Trends

AS above

Reviewer 2 Report

This manuscript describes the effects of different climate indices on the spring migration at two trapping station along the Polish Baltic coast.  Authors utilize various indices during different periods to reflect conditions in distinct phenological stages for wrens.  The paper is well-written and the results are extensively discussed.

Minor comments:

L15, L18, L34. "MOI1" should be corrected to "MOI".

L121. Figure 1 should be presented in higher resolution.

L124. The title should read "Study Site and Material", not "Study Siteand Material".

L143. Again, "MOI1" should be corrected to "MOI" throughout the entire manuscript.

L190, Table 1. Correct "Mean temperaturÄ™" and "Meant emperaturÄ™" to "Mean temperature".

L200-201. Please provide a broader explanation as to why certain years were excluded. It appears to be due to low mist-netting in Hell during those years. The authors should also add the number of mist-netting days for each station per year.

L241. Please add a reference to Table A5 in this section.

L237-238. The authors' intent is unclear. Are they performing separate model selection for each correlated variable, or including both variables in one global model and then selecting? If the latter, the analysis should be revised, as this is not the proper way to address this issue.

L271. "passageat" should be corrected to "passage". Also, insert a space in "Hel.Statistics".

L288. For clarity, consider add Bukowo-Kopań and Hel after panel sign A and B, respectively.

Also, please define what is meant by 'partial correlation' on y-axis. It's unclear whether a higher value signifies later arrival, and whether a negative value indicates earlier arrival.

Please consider also changing the variable names to reflect the short names used in the discussion, such as spring MOI, autumn MOI, winter NAO, etc., would be helpful. This change should be applied to the other tables as well.

L290. "(A )at" should be corrected to "(A) at".

L318. "autumnMOI1" should be corrected to "autumn MOI".

L382. Insert a space after "Table".

L364. Insert a space after "Hel".

L393. "Kopań(Fig. 4A)" should be corrected to "Kopań (Fig. 4A)".

L467. "[43],but" should be corrected to "[43], but".

Line 556. "Irends" should be corrected to "Trends".

Line 707, Table A3. If I understand correctly, each row represents a single linear regression analysis. However, it's unclear which variable is the dependent one. According to the description, 'year' is always the independent variable, and migration phenology variables are dependent. The table header titled 'parameter', indicating e.g. BK_AA, BK_5, etc., suggests these are independent variables, causing confusion.

Please also add an intercept for each model. The term "40 x beta (days)" is also unclear."
